# Optimization of Bobbin Tool Friction Stir Processing Parameters of AA1050 Using Response Surface Methodology

**DOI:** 10.3390/ma15196886

**Published:** 2022-10-04

**Authors:** Ibrahim Albaijan, Mohamed M. Z. Ahmed, Mohamed M. El-Sayed Seleman, Kamel Touileb, Mohamed I. A. Habba, Ramy A. Fouad

**Affiliations:** 1Mechanical Engineering Department, College of Engineering at Al Kharj, Prince Sattam Bin Abdulaziz University, Al Kharj 16273, Saudi Arabia; 2Department of Metallurgical and Materials Engineering, Faculty of Petroleum and Mining Engineering, Suez University, Suez 43512, Egypt; 3Mechanical Department, Faculty of Technology and Education, Suez University, Suez 43518, Egypt

**Keywords:** bobbin tool friction stir processing, processing parameters, AA1050, statistical modeling, Response Surface Method, Central Composite Design

## Abstract

The current research designed a statistical model for the bobbin tool friction stir processing (BT-FSP) of AA1050 aluminum alloy using the Response Surface Method (RSM). The analysis studied the influence of tool travel speeds of 100, 200, and 300 mm/min and different pin geometries (triangle, square, and cylindrical) at a constant tool rotation speed (RS) of 600 rpm on processing 8 mm thickness AA1050. The developed mathematical model optimizes the effect of the applied BT-FSP parameters on machine torque, processing zone (PZ) temperature, surface roughness, hardness values, and ultimate tensile strength (UTS). The experimental design is based on the Face Central Composite Design (FCCD), using linear and quadratic polynomial equations to develop the mathematical models. The results show that the proposed model adequately predicts the responses within the processing parameters, and the pin geometry is the most influential parameter during the BT-FSP of AA1050. The analysis of variance exhibit that the developed mathematical models can effectively predict the values of the machine torque, PZ temperature, surface roughness, hardness, and UTS with a confidence level of over 95% for the AA1050 BT-FSP. The optimization process shows that the optimum parameters to attain the highest mechanical properties in terms of hardness and tensile strength at the lowest surface roughness and machine torque are travel speed (TS) of 200 mm/min using cylindrical (Cy) pin geometry at the constant RS of 600 rpm. The PZ temperature of the processed specimens decreased with increasing TS at different pin geometries. Meanwhile, the surface roughness of the processed passes and machine torque increased with increasing the TS at different pin geometries. Increasing TS from 100 to 300 mm/min increases the hardness values of the processed materials using different pin geometries. The highest UTS of 79 MPa for the processed specimens was attained at the TS of 200 mm/min and RS of 600 rpm using the Cy pin geometry.

## 1. Introduction

AA1050 alloy has received special attention in many aerospace, electrical, and automotive industries due to its low density, high corrosion resistance, ease of formability, workability, and weldability [1,2]. Bobbin tool friction stir processing (BT-FSP) is a solid-state technique using a special tool design [3]. The bobbin tool (BT) name refers to the tool shape, consisting of two shoulders; one acts on the upper and one on the lower surface of the specimen under processing, connected by a pin. With this technique, the lower shoulder replaces the backing plate used in friction stir welding or processing (FSW/P) using a conventional tool (CT) design [4]. The BT design promotes symmetric heat generation during material processing [5,6] and eliminates root defects during friction stir welding (FSW) [7,8]. The stirring zone (SZ) temperature of the BT-FSW/P material is generated by the stirring tool action (pin and two shoulders) [9,10]. The processing zone (PZ) temperature during the BT-FSP depended directly on the rotational speed (RS) and inversely to travel speed (TS) [11,12,13]. Optimizing the processing temperature is critical to achieving a refined grain structure. Therefore, optimal RS and TS must be chosen to avoid defects in the stirring zone (SZ), such as the flash band [14,15,16]. In recent years, the friction stirring processing (FSP) technique using a CT design has been widely used for various aluminum alloys to improve mechanical properties and surface microstructures [17,18,19]. The Response Surface Method (RSM) analyzes the relations between many systems or processes’ factors and one or more response variables. By utilizing statistical methods such as the RSM, it is possible to maximize the production of a unique material by optimizing operational parameters. Recently, the RSM, in conjunction with an appropriate design of experiments (DOE), has become widely utilized for formulation optimization. In contrast to conventional methodologies, the RSM tool can determine the interaction between processing independent factors and modeling the system mathematically. In addition, it saves time and cost by reducing the number of trials [20,21]. Therefore, mathematical modeling using analysis of variance (ANOVA) was applied to study the most effective BT-FSW factors on the properties of the welded joints. Nasir et al. [22] optimized a mathematical model based on the face-centered composite design (FCCD) for 6 mm thickness AA1100 bobbin tool friction stir welding (BT-FSW) using different welding parameters in terms of rotation speeds (750, 850, and 950 rpm) and travel speeds (130, 150, and 170 mm/min) using a Cy pin with three flat feature. The ANOVA results showed that both the RS and TS significantly influence the average hardness and tensile strength of the AA1100 welded joints, and the optimum factors for maximum hardness and tensile strength are 130 mm/min and 950 rpm. Moreover, Zhao et al. [23] applied the FCCD method to study the effect of BT-FSW parameters on travel speeds (98, 150, 250, 350, and 402 mm/min) and rotation speeds (274, 300, 350, 400, and 425 rpm) using a Cy pin with different shoulder pinching gaps (5.40–5.80 mm) on the tensile strength and elongation properties of 6 mm thickness AA 2219-T87. The output results showed that the TS is the most significant parameter affecting the tensile and elongation properties compared to the rotation speed and pinching gap. In addition, the optimum BT-FSW parameters to obtain high efficiency joint are TS of 400 mm/min, RS of 345.5 rpm, and pinching gap of 5.66 mm. Ahmed et al. [24] applied a Taguchi statistical design to optimize the BT-FSW parameters of the 10 mm AA1050 lap joints using different pin geometries at different travel speeds (200, 400, and 600 mm/min) at a constant RS of 600 rpm. From the ANOVA results, welding temperature and tensile shear load were significantly affected by the TS, with a contribution of 73.64% and 84.77%, respectively. However, the hardness was mostly affected by the pin geometry, with a contribution of 71.51%. The available literature shows that the BT-FSW technique has been considered in many works to join aluminum alloys. This motivates using the BT design to process aluminum and its alloys to refine microstructure or produce aluminum matrix composites. Recently, only two works have been achieved for utilizing the BT-FSP to produce composite materials: Fuse et al. [25] produced AA6061/B_4_C Dual-sided composite, and Ahmed et al. [11] fabricated the AA1050/nano-silica fume composite. In addition, there is a lack of publication in the area of BT-FSP of aluminum alloys. Only one experimental study by Ahmed et al. [26] revealed that the BT-FSP parameters of 200 mm/min and 600 rpm for AA1050 deliver the highest ultimate tensile strength of 79 MPa and hardness of 36.2 HV with an enhancement of 33.8% and 20.6% compared to the base material. The RSM is a valuable method to determine effective process factors and the interaction between them. The RSM has been applied in many manufacturing sectors, such as casting [27,28], fusion welding [21,29], and powder metallurgy [30,31]. Based on the available literature, some investigations involved the RSM in the FSW and FSP using a conventional tool [20,32]. Moreover, no studies were found applying the RSM to optimize the BT-FSP parameters for AA1050 alloy. Therefore, this study aimed to investigate the impacts of different pin geometries of Tr, Sq, and Cy and travel speeds of 100, 200, and 300 mm/min on the temperature of the PZ, surface roughness, and the mechanical properties in terms of hardness values and ultimate tensile strength of the BT-FSP of 8 mm thickness AA1050 alloy based on the Response Surface Method (RSM).

## 2. Materials and Methods

### 2.1. Materials and Processing

The as-received material used in the optimization experimentation was an 8 mm thick sheet of AA1050 aluminum alloy; the nominal composition and mechanical properties are listed in Table 1. Rectangular plates were cut from the supplied sheet with dimensions of 150 mm long and 75 mm wide to produce AA1050 processing paths using the BT-FSP technique. The cut plates were cleaned by a steel brush on the upper and lower surfaces of the plates. After cleaning with acetone, the AA1050 plate was clamped on the worktable of the full-automatic FSW/P machine (Model: EG-1 FSW/FSP machine, Suez and Sinai Metallurgical and Materials Research Center of Scientific Excellence, Suez University, Suez, Egypt) [33,34] with a special fixture (Figure 1a) for the BT-FSP purposes (Figure 1b). Figure 2a–c present the dimensions of the BT designed for the 8 mm AA1050 BT-FSP purposes. The bobbin tools were manufactured from cold-worked H13 tool steel and heat-treated to achieve a 52 HRC hardness.

The BT-FSP experiments of the AA1050 plates were carried out at a constant RS of 600 rpm using different pin geometries at different travel speeds. Table 2 lists the BT-FSP variables and their levels. In addition, the plunge depth of the top and bottom shoulders of the BT-FSP was kept constant (0.07 mm) during all the experiments. It is noteworthy that for all applied variables of BT-FSP, each sample was produced three times using the same variables.

### 2.2. Material Characterization

The temperatures of the PZ for the produced AA1050 processing paths were measured using an infrared thermometer (Quicktemp 860-T3, Testo Company, Berlin, Germany). The BT-FSP torque was recorded during the BT-FSP of AA1050 with the FSW/P machine for the applied processing variables. The torque value is a good indicator of the material’s resistance to stirring around the pin. The surface roughness of the BT-FSPed paths was measured using Posi-Tector Surface Profile Gages (Ogdensburg, NY, USA). The AA1050 as-received material and the processed paths were cut perpendicular to the BT-FSP direction for the hardness and tensile tests, as shown in Figure 3a. The hardness measurement of the PZ was performed using a Vickers hardness tester (Model: HWDV-75, TTS Unlimited, Osaka, Japan) with a 300 g load at 15 s dwell time. For each processing path, the average hardness was calculated for not less than 30 readings. Figure 3b shows the dimensions of the tensile test specimen according to ASTM-E8. The tensile test for the as-received material and the processed paths were evaluated using a universal testing machine (model: WDW-300D Testing Machine, 30-ton, Dongguan, China). The tensile test was carried out for three samples processed under the same conditions. After the tensile test, the fracture surfaces of the failed specimens were investigated using a scanning electron microscope (SEM, Thermo-Scientific, Quattro S, Waltham, MA, USA). 

### 2.3. Mathematical Modeling

The Design of Experiments (DOE) program is a set of techniques that examine the influence of different parameters for the BT-FSP on the results of a controlled experiment. The first step is to identify the independent variables or factors affecting the process and examine their impact on a dependent variable or response. The Design-Expert software program was applied to analyze the experimental data and plot the reaction surface. The ANOVA and the RSM were used to estimate the statistical variables. The degree of fitting of the experimental results to the polynomial model equation was expressed by the coefficient of determination R^2^. The F-test was used to estimate the significance of each term in the polynomial equation within a 95% confidence interval. According to the illustrated design, the different conditions of the BT-FSP process were estimated using a second-order polynomial, through which a correlation between the examined factors and the response (machine torque, surface roughness of the processed paths, the PZ temperature, hardness, and UTS) is generated. Accordingly, the optimal conditions were estimated using a second-order polynomial function, which generates a correlation between the factors under study and the response. The general form of the equation is given in Equation (1) [35,36]:(1)$$Y=\beta_{o}+\sum_{i=1}^{k}\beta_{i}\;X_{i}+\sum_{i=1}^{k}\beta_{ii}\;X^{2}{}_{i}+\sum_{i=1}^{k}\;\sum_{i<j}^{k}\;\beta_{ij}\;X_{i}X_{j}+\varepsilon$$
where *Y*, *X_i_*, *β*_o_, *β_i_*, *β_ii_*, *β_ij_*, *k*, and *ε* are the predicted response, independent input variable, constant-coefficient, linear coefficient, quadratic coefficient, interaction coefficient, represent the number of process parameters, and residual error, respectively. Finally, a confirmatory test was performed under optimal conditions for validation regarding the error between experimental and theoretical results. The Design-Expert V.13 computer software programming with the RSM technique was used to build the matrix of input parameters. In this study, a data analysis based on the FCCD method (L9 orthogonal array) was performed to estimate the significant factors of the 8 mm AA1050 BT-FSP. A complete design matrix with observed response variables is listed in Table 3.

## 3. Results and Discussion

### 3.1. Development of Mathematical Models

Commercial statistical software was used to analyze the measured responses (PZ temperature, recorded machine torque, the surface roughness of processed paths, hardness, and UTS) and develop best-fit mathematical models. On the other hand, the adequacy of models was tested using the ANOVA technique.

The mathematical regression model uses the stepwise regression method. First, the mathematical regression model eliminates the insignificant terms and computes the regression coefficients until the significant terms, and the regression model terms meet the requirements of the regression model. The relational expressions for all the BT-FSP variables of the AA1050 sheet within TS in the range of 100 to 300 mm/min were obtained as Equations (2)–(6), which show the relationship between input variables and the responses; machine torque, the PZ temperature, the surface roughness of processed paths, hardness, and UTS. Finally, ANOVA was used to determine the significance and appropriateness of the mathematical regression model. Table 4, Table 5, Table 6, Table 7 and Table 8 represent the ANOVA analysis of the recorded machine torque, PZ temperature, surface roughness of the processed zone, hardness values, and UTS mathematical model. The values of “Prob > F” less than 0.0500 indicate mathematical model terms are significant, and a value greater than 0.1000 indicate the model terms are not significant [37]. Table 4, Table 5, Table 6, Table 7 and Table 8 show the result of ANOVA for the recorded machine torque model, PZ temperature model, surface roughness in processing paths, hardness value, and the UTS models, respectively, and the F-values of the mathematical models are 29.82, 349.14, 93.55, 307.55, and 650.71. The probability of P-values is less than 0.05; in other words, these statistical models are significant. According to Table 4 and Table 5, A, B, AB, A^2,^ and B^2^ are the most important recorded machine torque and PZ temperature model factors. From Table 6, A and B are the most important factors of the surface roughness of processed paths; A, B, AB, A^2^, B^2^, A^2^B, and AB^2^ could affect hardness values, and the UTS model is shown in Table 7 and Table 8. In addition, Table 4, Table 5, Table 6, Table 7 and Table 8 also represented the other adequacy R^2^, adjusted R^2^, predicted R^2^, and adequate precision. All adequacy R^2^ values are logically consistent and indicate significant relationships. The reasonable accuracy ratios (signal-to-noise ratio) are higher than 4 in all cases, indicating an adequate signal for all statistical models used in this research.
Machine Torque (N·m) = +37.66667 − 0.003333 * × TS + 29.83333 × Pin geometry + 7.53033 × 10^−17^ × TS × Pin geometry + 0.000100 × TS^2^ − 8.50000 × Pin geomery^2^(2)
PZ Temperature (°C) = +342.22222 − 0.591667 × TS + 55.5 × Pin geometry − 0.0175 × TS × Pin geometry + 0.000867 × TS^2^ − 8.33333 × Pin geometry^2^(3)
Surface Roughness (µm) = +247.98889 + 0.194833 × TS − 54.75000 × Pin geometry(4)
Hardness (HV) = +11.92222 + 0.188333 × TS + 5.28333 × Pin geometry − 0.049 × TS × Pin geometry − 0.000373 × TS^2^ − 0.383333 × Pin geometry^2^ + 0.000065 × TS^2^ × Pin geometry + 0.006 × TS × Pin geometry^2^(5)
UTS (MPa) = +23.44444 + 0.491667 × TS + 20.16667 × Pin geometry − 0.12 × TS × Pin geometry − 0.001417 × TS^2^ − 2.16667 × Pin geometry^2^ + 0.0003 × TS^2^ × Pin geometry + 0.005 × TS × Pin geometry^2^(6)

### 3.2. Effect of Pin Geometry and Travel Speed on Response Variables

Figure 4a–e show the 3D response surface influence of TS and pin geometry on the machine torque, PZ temperature, surface roughness, hardness values, and UTS, respectively. The main objective was to control and reduce all response surface variables. Figure 4a represents the influence of the TS and pin geometry on the recorded machine torque. It is clear from Figure 4a that the torque is more affected by the TS than the pin geometry. Moreover, using the Cy pin geometry, the torque attains the minimum value (52 N·m) at the lowest TS (100 mm/min). The maximum torque value gains is 73 N·m at the highest TS of 300 mm/min using the Sq pin geometry. In addition, the machine torque increases with increasing the TS at all the applied pin geometries. Figure 4b shows the influence of the TS and pin geometry on the PZ temperature. It can be remarked that the PZ temperature reaches the lowest value of 286 °C at the highest processing TS of 300 mm/min using the Tr pin geometry, and the highest temperature value of 377 °C is generated using the Cy pin geometry at the lowest Ts of 100 mm/min. It can be noted that the PZ temperature is more affected by the pin geometry than the TS, as shown in Figure 4b. 

Figure 4c reflects the influence of TS and pin geometry on the surface roughness of the processed paths. It was found that the pin geometry influences the surface roughness more than the TS. Furthermore, the surface roughness increases with increasing the TS value. The lowest value of surface roughness (108.2 µm) was obtained for the processed specimen at 100 mm/min TS using the Cy pin geometry. Hardness values were measured on the cross-section areas of the produced BT-FSP specimens. The highest hardness value was obtained at the processing TS of 200 mm/min, while the lowest hardness value was obtained at the lowest TS of 100 mm/min using all pin geometries. Figure 4d shows the interaction of the processing variables with the responses and the 3D surface of the hardness results. Increasing TS from 100 to 300 mm/min generally increases the hardness values for the applied pin geometries. This hardness trend is probably due to decreased PZ temperature (heat input) with increasing TS [38]. 

The main objective of the tensile test was to evaluate the strength and plasticity of the BT-FSP materials and to study the influence of processing variables on the performance of the produced specimens. The maximum UTS was obtained at the mean TS of 200 mm/min for all the pin geometries, and the minimum UTS was obtained at the highest TS of 300 mm/min, as given in Figure 4e. Compared to the as-received material, this improvement is probably due to grain refining through dynamic recrystallization in the PZ [39,40]. The fine grain structure increases the UTS of the processed materials due to the impediment to dislocation movement [41,42,43]. In addition, the analysis and parametric influence examined in ANOVA can be seen in Table 8, and the 3D surface (interaction) plots are presented in Figure 4e. The results show that pin geometries have more influence on the UTS than the TS of the processed AA1050.

Figure 5 illustrates the fracture surfaces for the processed specimens after the tensile test for the AA1050 as-received material (Figure 5a) and the processed samples using the three-pin geometries processed at a constant RS of 600 rpm and the travel speeds of 50 mm/min (Figure 5b,d,f) and 300 mm/min (Figure 5c,e,g). The SEM image of AA1050 base material (BM) aluminum alloy shows different dimples (large and small dimples), this fractography indicating ductile fracture mode, as presented in Figure 5a. Figure 5b–g also contain equiaxed deep and shallow dimples smaller than those detected for the as-received material. These different dimples indicate grain refining in the PZ caused by the dynamic recrystallization combined with the applied BT-FSP variables. Moreover, the processed materials using pin geometries of Tr, Sq, and Cy at a 50 mm/min (high heat input) show large and deep dimples, as shown in Figure 5b,d,f, respectively, compared to those processed at 300 mm/min (low heat input), as shown in Figure 5c,e,g. Finally, the fracture surface images of the processed specimen at 200 mm/min TS using Cy pin were dominated by equiaxed, homogeneous, and uniform smaller dimples (Figure 6a–c) compared to other processed specimens, indicating higher grain refinement. These characteristics were confirmed with the obtained results of hardness and UTS.

### 3.3. Validation of the Developed Mathematical Models

Table 9 demonstrates the relationship between the predicted and actual values of the machine torque, PZ temperature, hardness values, and UTS, respectively. These tables summarize the experiment’s parameters, the predicted values, the actual experimental values, and the error percentages. In order to validate the empirical equations of the developed mathematical model response surface obtained from multiple regression analysis, three confirmation experiments were performed with randomly selected BT-FSP variables within and without the range for which the empirical equations are obtained (Equations (2)–(6)). The percentage error between actual and predicted values can also be calculated via Equation (7) [44]. As shown in Table 9, all error values are in the range of engineering errors and are accepted in the industry. 

Figure 7a–e shows the relevance between the actual and predicted values of machine torque, PZ temperature, surface roughness, hardness, and UTS, respectively. These figures also mention that the mathematical models developed are appropriate, and the predicted results agree well with the experimental results. Figure 7a revealed the recorded machine torque relationship between the predicted and actual value; meanwhile, these relations revealed the excellent agreement of actual and predicted values. In order to check the adequacy of the developed model, the comparison graph was created between the predicted processing temperature values obtained from the mathematical model and the experimental values shown in Figure 7b.

The connection between predicted and actual values for surface roughness is presented in Figure 7c. Moreover, this Figure demonstrates that the predicted values by the surface roughness model are in good agreement with actual values. In addition, Figure 7d shows the empirical relationship between experimented hardness and predicted hardness values, and this relationship is a good agreement. In order to check the adequacy of the developed model, the comparison curve was drawn between the predicted tensile strength values obtained from the model and the experimental values (Figure 7e). Figure 7e shows a good agreement between predicted and experimental values, thus, suggesting the adequacy of the mathematical regression model.
(7)$$Percentage\;error\;\left(E\%\right)=\left|\frac{Actual\;value-Predicted\;value}{Predicted\;value}\right|*100$$

## 4. Conclusions

In this study, statistical models for the bobbin tool friction stir processing (BT-FSP) of AA1050 using the Response Surface Method (RSM) were designed and analyzed. The effect of two main input parameters was investigated, tool travel speeds of 100, 200, and 300 mm/min and tool pin geometries (triangle, square, and cylindrical) at a constant tool rotation speed of 600 rpm. The following conclusions can be outlined: The RSM based on face center composed design (FCCD) can be used effectively to analyze the cause and effect of the BT-FSP variables in the response. RSM was also used to draw contour diagrams for different responses to show the interaction effects of various processing parameters;The analysis of variance exhibited that the developed mathematical models can effectively predict the values of machine torque, PZ temperature, surface roughness, hardness, and UTS with a confidence level of over 95% for the AA1050 BT-FSP;The PZ temperature of the processed specimens decreased with increasing TS at different pin geometries. Moreover, the surface roughness of the processed paths and machine torque increased with increasing the TS at different pin geometries;Increasing TS from 100 to 300 mm/min increases the hardness values for the applied pin geometries. The best result was obtained from Cy pin geometry;The higher UTS obtained for the processed specimens produced at the TS of 200 mm/min and RS of 600 rpm using the Cy pin geometry.

## Figures and Tables

**Figure 1 materials-15-06886-f001:**
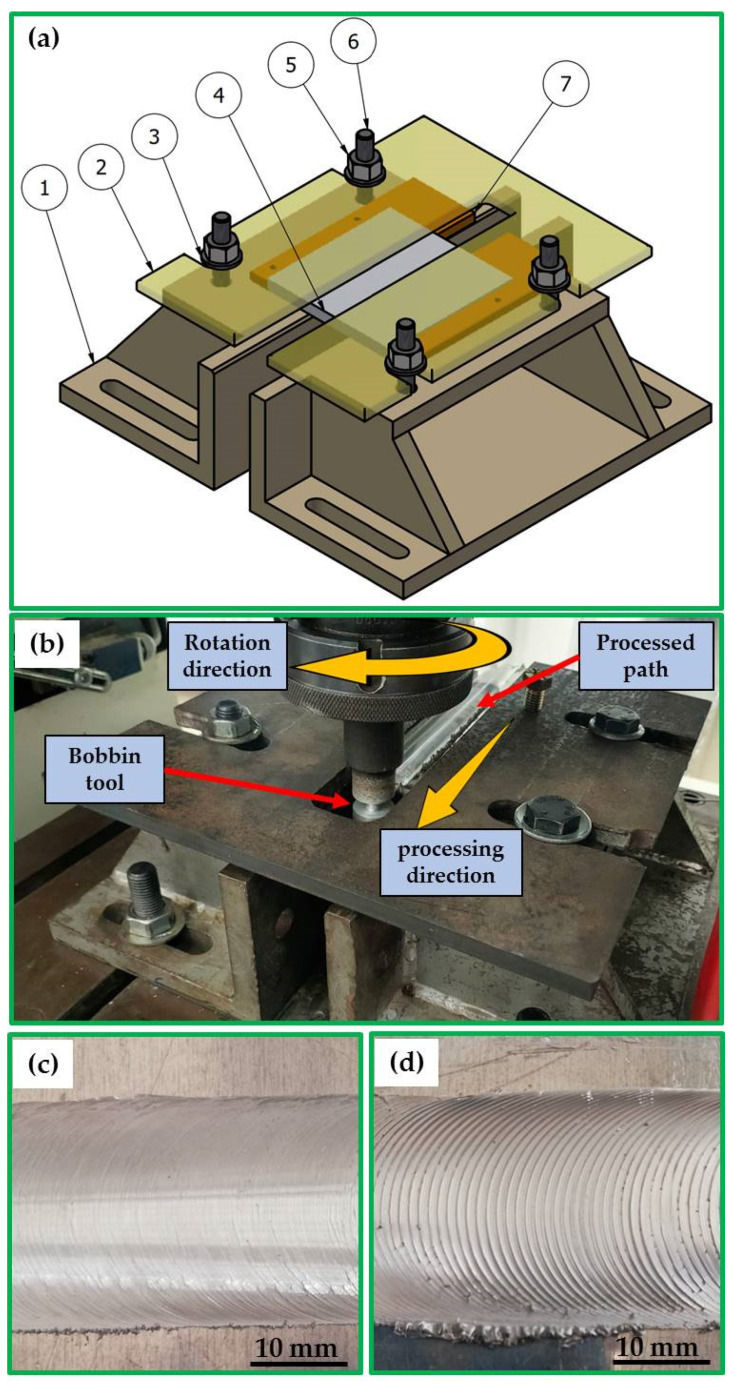
(**a**) Three-dimensional drawing of the fixture setup, (1) fixture base, (2) covering plate, (3) washer, (4) AA1050 plate, (5) nut, (6) bolt, and (7) fixation plate; (**b**) photo image of the BT-FSP of AA1050, (**c**) and (**d**) the top surfaces of the processed specimens at traveled speeds of 100 and 300 mm/min, respectively, using the Cy pin geometry.

**Figure 2 materials-15-06886-f002:**
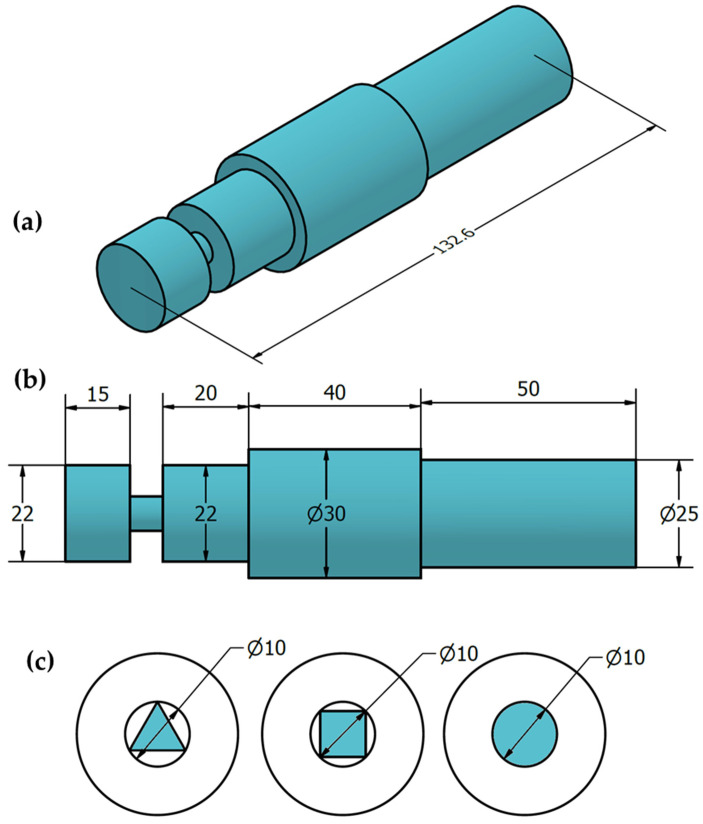
(**a**) Three-dimensional drawing and (**b**) dimensions of the applied BT, and (**c**) the used different pin geometries. (Unit: mm).

**Figure 3 materials-15-06886-f003:**
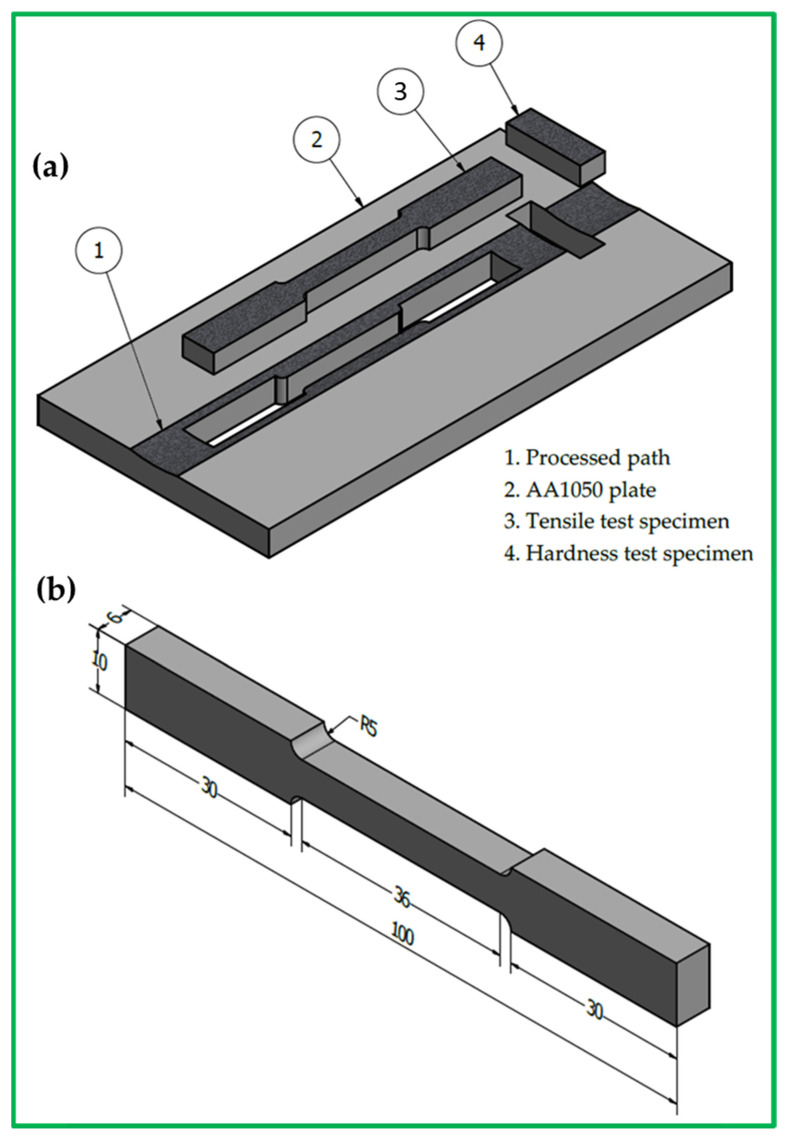
Three-dimensional drawings present (**a**) the location of tensile and hardness test specimens from the processed path and (**b**) the tensile specimen dimensions (all dimensions in mm).

**Figure 4 materials-15-06886-f004:**
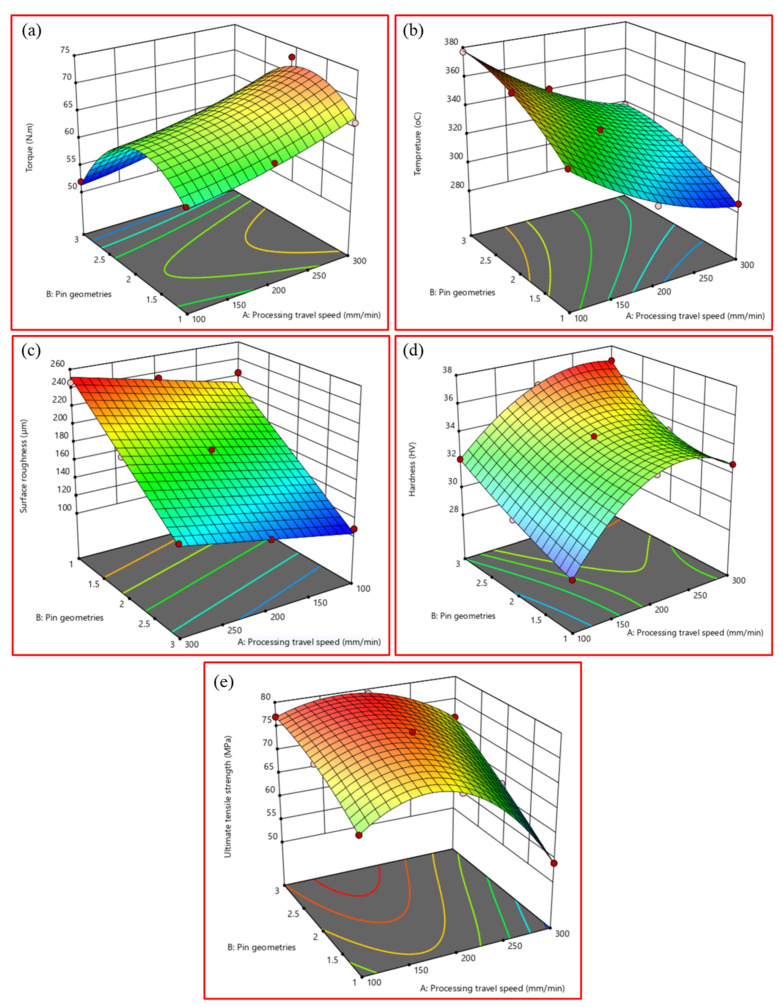
Three-dimensional contour diagrams for the (**a**) recorded machine torque, (**b**) PZ temperature, (**c**) surface roughness, (**d**) hardness, and (**e**) tensile strength.

**Figure 5 materials-15-06886-f005:**
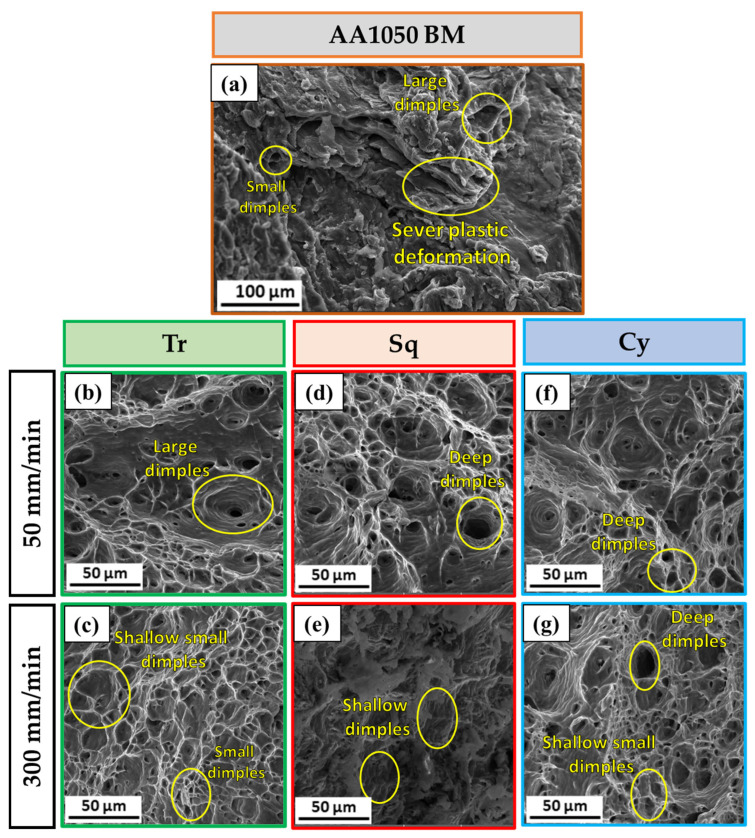
SEM images of the fracture surfaces of the failed specimens after the tensile test for (**a**) AA1050 BM and (**b**–**f**) and (**c**–**g**) for the processed materials using the three-pin geometries at the BT-FSP variables of TS of 50 mm/min and 300 mm/min, respectively.

**Figure 6 materials-15-06886-f006:**
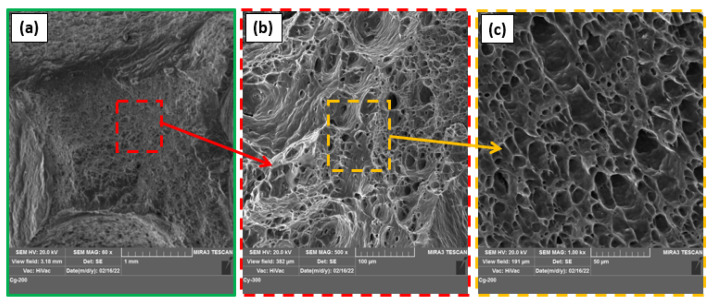
(**a**–**c**) SEM images of the fracture surfaces for the BT friction stir processed at 200 mm/min using Cy pin geometry at different magnifications.

**Figure 7 materials-15-06886-f007:**
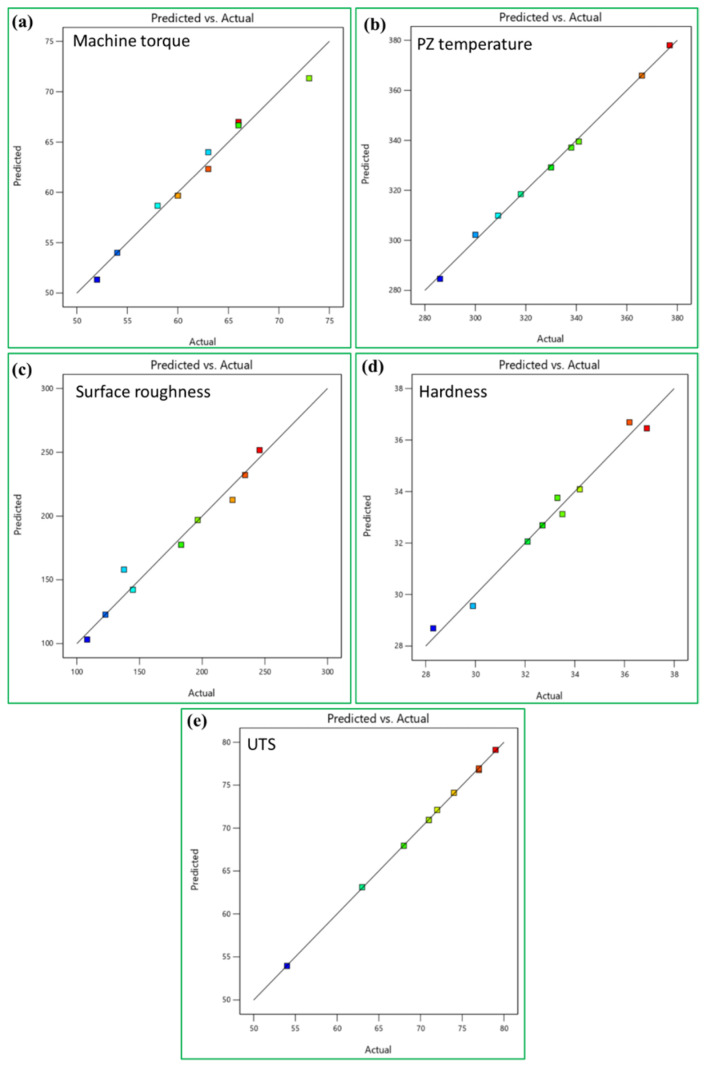
Correlation diagram between the actual and predicted values of the (**a**) recorded machine torque model, (**b**) PZ temperature, (**c**) surface roughness, (**d**) hardness, and (**e**) tensile strength.

**Table 1 materials-15-06886-t001:** The nominal composition and mechanical properties of the as-received AA1050 sheets.

Chemical Composition									
Element	Si	Zn	Mn	Mg	Cr	Fe	Cu	Ti	Al
Wt. %	0.0889	0.0019	0.0002	0.0030	0.0012	0.257	0.0031	0.0139	Bal.
**Mechanical Properties**									
**Property**	**UTS (MPa)**					**HV**			
AA1050	59 ± 2					31 ± 2			

**Table 2 materials-15-06886-t002:** The BT-FSP variables and their levels.

BT-FSP Variables	Units	Levels		
		Level 1	Level 2	Level 3
Travel speed (TS)	mm/min	100	200	300
Tool geometry	-	Triangle (Tr)	Square (Sq)	Cylindrical (Cy)

**Table 3 materials-15-06886-t003:** Design matrix for 8 mm thickness AA1050 BT-FSP.

Run	Input Processing Variables		Response Processing Variables				
	A: Travel Speed, mm/min	B: Pin Geometries	Temperature, °C	Torque, N·m	Surface Roughness, µm	HV	UTS, MPa
1	300	1 (Tr)	286	66	245.8	32.7	54
2	100	1 (Tr)	338	60	224.2	28.3	68
3	300	3 (Cy)	318	58	144.7	36.9	71
4	200	3 (Cy)	341	54	122.8	36.2	79
5	300	2 (Sq)	309	73	196.4	33.3	63
6	100	3 (Cy)	377	52	108.2	32.1	77
7	100	2 (Sq)	366	63	137.6	29.9	74
8	200	2 (Sq)	330	66	183.2	34.2	77
9	200	1 (Tr)	300	63	234.2	33.5	72

**Table 4 materials-15-06886-t004:** ANOVA for machine torque of the AA1050 BT-FSP specimens.

Source	Sum of Squares	df	Mean Squares	F-Value	*p*-Value
Model	331.33	5	66.27	29.82	0.0092
A-Travel speed	80.67	1	80.67	36.30	0.0092
B-Pin geometry	104.17	1	104.17	46.87	0.0064
AB	0.0000	1	0.0000	0.0000	1.0000
A^2^	2.00	1	2.00	0.9000	0.4128
B^2^	144.50	1	144.50	65.03	0.0040
Residual	6.67	3	2.22	-	-
Cor Total	338.00	8	-	-	-
Std. Dev.	1.4	R^2^		0.9803	
Mean	61.67	Adjusted R^2^		0.9474	
CV %	2.42	Predicted R^2^		0.7736	
		Adeq precision		16.4317	

**Table 5 materials-15-06886-t005:** ANOVA for PZ temperatures of AA1050 BT-FSP specimens.

Source	Sum of Squares	df	Mean Squares	F-Value	*p*-Value
Model	7096.03	5	1419.21	349.14	0.0002
A-Travel speed	4704.00	1	4704.00	1157.25	<0.0001
B-Pin geometry	2090.67	1	2090.67	514.33	0.0002
AB	12.25	1	12.25	3.01	0.1810
A^2^	150.22	1	150.22	36.96	0.0089
B^2^	138.89	1	138.89	34.17	0.0100
Residual	12.19	3	4.06	-	-
Cor Total	7108.22	8	-	-	-
Std. Dev.	2.02	R^2^		0.9983	
Mean	329.44	Adjusted R^2^		0.9954	
CV %	0.6120	Predicted R^2^		0.9800	
		Adeq precision		56.6973	

**Table 6 materials-15-06886-t006:** ANOVA for surface roughness of AA1050 BT-FSP specimens.

Source	Sum of Squares	df	Mean Squares	F-Value	*p*-Value
Model	20,262.98	2	10,131.49	93.55	<0.0001
A-Travel speed	2277.60	1	2277.60	21.03	0.0037
B-Pin geometry	17,985.38	1	17,985.38	166.07	<0.0001
Residual	649.81	6	108.30	-	-
Cor Total	20,912.78	8	-	-	-
Std. Dev.	10.41	R^2^		0.9689	
Mean	177.46	Adjusted R^2^		0.9586	
CV %	5.86	Predicted R^2^		0.9290	
		Adeq precision		24.710	

**Table 7 materials-15-06886-t007:** ANOVA for the hardness values of AA1050 BT-FSP specimens.

Source	Sum of Squares	df	Mean Squares	F-Value	*p*-Value
Model	59.80	7	8.54	307.55	0.0439
A-Travel speed	5.78	1	5.78	208.08	0.0441
B-Pin geometry	3.65	1	3.65	131.22	0.0554
AB	0.040	1	0.040	1.44	0.4423
A^2^	11.84	1	11.84	426.32	0.0308
B^2^	1.33	1	1.33	48.02	0.0912
A^2^B	0.5633	1	0.5633	20.28	0.1391
AB^2^	0.48	1	0.48	17.28	0.1503
Residual	0.0278	1	0.0278	-	-
Cor Total	59.83	8	-	-	-
Std. Dev.	0.1667	R^2^		0.9995	
Mean	33.01	Adjusted R^2^		0.9963	
CV %	0.5049	Predicted R^2^		0.9154	
		Adeq precision		54.7301	

**Table 8 materials-15-06886-t008:** ANOVA for UTS of AA1050 BT-FSP specimens.

Source	Sum of Squares	df	Mean Squares	F-Value	*p*-Value	
Model	506.11	7	72.30	650.71	0.0302	Significant
A-Travel speed	60.50	1	60.50	544.50	0.0273	
B-Pin geometry	24.50	1	24.50	220.50	0.0428	
AB	16.00	1	16.00	144.00	0.0529	
A^2^	133.39	1	133.39	1200.50	0.0184	
B^2^	2.72	1	2.72	24.50	0.1269	
A^2^B	12.00	1	12.00	108.00	0.0611	
AB^2^	0.3333	1	0.3333	3.00	0.3333	
Residual	0.1111	1	0.1111	-	-	
Cor Total	506.22	8	-	-	-	
Std. Dev.	0.3333	R^2^		0.9998		
Mean	70.56	Adjusted R^2^		0.9982		
CV %	0.4724	Predicted R^2^		0.96		
		Adeq precision		80.0798		

**Table 9 materials-15-06886-t009:** Confirmation of test results for mathematical models.

Exp. No	TS, mm/min	Pin Geometry		Torque, N·m	Tem, °C	Surface Roughness, µm	Hardness, HV	UTS, MPa
			Actual	58	380	213.5	24.8	59
1.	50	Tr	Predicted	59.1	361.1	203	23.32	57.49
			|Error|, %	1.9	5.23	5.17	6.35	2.63
			Actual	62	330	229.23	30.86	73
2.	150	Tr	Predicted	60.75	317.52	222.46	31.69	72.81
			|Error|, %	2.06	3.93	3.04	2.62	0.26
			Actual	63.58	281.47	238.79	34.2	63.85
3.	250	Tr	Predicted	64.42	291.28	241.95	33.91	65.3
			|Error|, %	1.3	3.37	1.31	0.86	2.22

## Data Availability

Data will be available upon request through the corresponding author.

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
