# Peer review of "Optimization of Bobbin Tool Friction Stir Processing Parameters of AA1050 Using Response Surface Methodology"

_materials, 2022, doi:10.3390/ma15196886_

Round 1

Author Response

Response to comments of Reviewer (1)

Dear Reviewer

We do appreciate your positive feedback and the constructive comments you have provided that improved our manuscript significantly.

Comment #1:

Abstract: Please note that abbreviations must be spelled in full the first time they appear, e.g.:

"PZ" on line 28, "TS" on line 31, "Cy" on line 35, "RS" on line 36 etc.

Response:

Many thanks for the comment.

All abbreviations and their related full words in the abstract section and the manuscript have been revised and clarified.

Comment #2:

Introduction: Please declare the reason why the Response Surface Method (RSM) was selected over other methods, and what are the advantages of this method?

Response:

Many thanks for the comment.

The reason for selecting the Response Surface Method (RSM) and its advantages compared to the other methods have been briefly explained in the introduction section of the revised manuscript with adding the relevant references [20-21].

Comment #3:

Materials and Methods: Please check the caption of Figure 1. (a), that the number identifier must be the same as in the figure.

Response:

Many thanks for the comment.

Figure 1a and its number identifiers have been modified in the revised manuscript.

Comment #4:

Please write the formula comer symbol correctly on line 169.

Response:

Many thanks for the comment.

The formula and its symbols have been corrected.

Comment #5:

Results and Discussion: It is recommended to keep the 3D response surface, delete the 2D contours, and merge the 3D response surface influence of TS and pin geometry on the machine torque, PZ temperature, surface roughness, hardness values, and UTS for Figures 4-8.

Response:

Many thanks for the comment.

The 2D contour maps have been deleted, and the 3D response surface graphs have been merged into one Figure (Figure 4a-e) instead of Figures 4-8.

Comment #6:

Please add the clear labels of Figure 9 to distinguish the size of the dimples.

Response:

Many thanks for the comment.

Figure 6 has been modified in the revised manuscript.

Comment #7:

It is recommended to merge the Figures 11-15.

Response:

Many thanks for the comment.

Figures 11-15 have been merged in one Figure (Figure 7) in the revised manuscript.

Reviewer 2 Report

This manuscript studies the design a statistical model for bobbin tool friction stir processing of AA1050 aluminum alloy using the Response Surface Method (RSM). The analysis studied the influence of tool travel speeds of 100, 200, and 300 mm/min and different pin geometries at a constant tool rotation speed of 600 rpm on processing 8 mm thickness AA1050. The developed mathematical model optimizes the effect of the applied parameters on machine torque, processing zone temperature, surface roughness, hardness values, and ultimate tensile strength.

Global Comments

Interesting document but needs to correct some minor mistakes.

Conclusions can be improved.

The scientific robustness of the proposed method is adequate, and the results and the conclusions can be improved.

So, the manuscript needs minor improvement.

 Comments - others

 Abstract:

Page 1, Line 17:

  Include: travel speed (TS)

 Page 1, Line 22:

  Include: ultimate tensile strength (UTS)

 3. Results and Discussion:

Page 9-10, Line 230:

     Table 7. ANOVA for the hardness values of AA1050 BT-FSP specimens. Table Incorrectly showed.

Page 17, Line 301, 302, 303, 304, Figure 9

     Complete the legend, including the reference to Figure 9a).

     Include the designation BM - Base material, 

Page 17, Line 287

Page 19, Line 341, Table 9,

      Why the analysis is being done for a travel speed of 50mm/min? The main objective of this study is to analyze the influence of tool travel speeds of 100, 200, and 300 mm/min. So?

4. Conclusions

    Improving the conclusions

Author Response

Response to comments of Reviewer (2)

This manuscript studies the design a statistical model for bobbin tool friction stir processing of AA1050 aluminum alloy using the Response Surface Method (RSM). The analysis studied the influence of tool travel speeds of 100, 200, and 300 mm/min and different pin geometries at a constant tool rotation speed of 600 rpm on processing 8 mm thickness AA1050. The developed mathematical model optimizes the effect of the applied parameters on machine torque, processing zone temperature, surface roughness, hardness values, and ultimate tensile strength.

Global Comments

Comment #1:

Interesting  document  but  needs to correct some minor mistakes. Conclusions can be improved. The scientific robustness of the proposed method is adequate, and the results and the conclusions can be improved. So, the manuscript needs minor improvement.

Response:

Many thanks for the comment.

The manuscript has been revised with improving the discussion and conclusions

 Comments - others

Abstract:

Comment #2:

Page 1, Line 17: Include: travel speed (TS)

Response:

Many thanks for the comment.

The abbreviation of travel speed (TS) has been added in the revised manuscript.

Comment #3:

 Page 1, Line 22: Include: ultimate tensile strength (UTS)

Response:

Many thanks for the comment.

The abbreviation UTS has been identified in the revised manuscript.

Results and Discussion:

Comment #4:

Page 9-10, Line 230: Table 7. ANOVA for the hardness values of AA1050 BT-FSP specimens. Table Incorrectly showed.

Response:

Many thanks for the comment.

The show of Table 7 has been modified in the revised manuscript.

Comment #5:

Page 17, Line 301, 302, 303, 304, Figure 9: Complete the legend, including the reference to Figure 9a). Include the designation BM - Base material,

Response:

Many thanks for the comment.

The legend of Figure 9 (Figure 5 in the revised manuscript) has been modified.

Comment #6:

Page 17, Line 287 and Page 19, Line 341, Table 9: Why the analysis is being done for a travel speed of 50 mm/min? The main objective of this study is to analyze the influence of tool travel speeds of 100, 200, and 300 mm/min. So?

Response:

Many thanks for the comment.

The AA1050 alloy was processed using BT-FSP at a travel speed of 50 mm/min to verify the effectiveness of the equations (Equations 2-6) extracted from the ANOVA analysis and to test their accuracy outside the range of the experiments studied. And the prediction error of the results was within the allowable range, as shown in Table 9 in the revised manuscript. Hence, the fracture surface of the processed specimens at a travel speed of 50 mm/min was studied, as shown in Figure 5 in the revised manuscript.

Conclusions

Comment #7:

Improving the conclusions

Response:

Many thanks for the comment.

The conclusion section has been improved.

Reviewer 3 Report

The article describes the application of methods of mathematical planning of the experiment, in particular the methodology of the surface of the response, to the optimization analysis of friction processing of aluminum alloy. The regression dependencies and their results are important and interesting for technologists in machines building.

The text of the manuscript meets the requirements of MDPI. I am solidarity with the desire of the authors to publish this article in the journal "Materials".

Remark

Where is the position 3 in Fig. 3a?

 Conclusion

The article can be published after minor corrections.

Author Response

Response to comments of Reviewer (3)

The article describes the application of methods of mathematical planning of the experiment, in particular the methodology of the surface of the response, to the optimization analysis of friction processing of aluminum alloy. The regression dependencies and their results are important and interesting for technologists in machines building.

The text of the manuscript meets the requirements of MDPI. I am solidarity with the desire of the authors to publish this article in the journal "Materials".

Comment #1:

Where is the position 3 in Fig. 3a?

Response:

Many thanks for the comment.

Figure 3a has been corrected in the revised manuscript.

Reviewer 4 Report

The manuscript is well written, well-focused, and can be an interesting contribution to the modern industry. But there are some issues to improve.

·       The key contribution of this paper should be concentrated and highlighted. I suggest the authors explain better and clearly state the benefits of their research and their result.

·       Clearly describe the novelty of the research and add information about the practical significance of the work.

·       No pictures of the surface were shown after machining.

·       Information on surface roughness parameters was not provided (which parameter is described as surface roughness), and also the surface profile or surface roughness (3D) should be shown and described.

·       No information about experiment run repetition.

·       Where can we find the optimization described in the abstract? The optimization process should give information about minimalization or maximization value of the response variables.

Author Response

Response to comments of Reviewer (4)

The manuscript is well written, well-focused, and can be an interesting contribution to the modern industry. But there are some issues to improve.

Comment #1:

The key contribution of this paper should be concentrated and highlighted. I suggest the authors explain better and clearly state the benefits of their research and their result.

Response:

Many thanks for the comment.

The benefits of the research have been clarified in the revised manuscript in the introduction section and supported with the related references.

Comment #2:

Clearly describe the novelty of the research and add information about the practical significance of the work.

Response:

Many thanks for the comment.

The novelty and the significance of this work are described in the revised manuscript (introduction section)

Comment #3:

No pictures of the surface were shown after machining.

Response:

Many thanks for the comment.

The figures of the processing surface after the BT-FSP (Figure 1c,d) have been added to the revised manuscript.

Comment #4:

Information on surface roughness parameters was not provided (which parameter is described as surface roughness), and also the surface profile or surface roughness (3D) should be shown and described.

Response:

Many thanks for the comment.

This point has been treated in the revised manuscript.

Comment #5:

No information about experiment run repetition.

Response:

Many thanks for the comment.

The information about the repetition of the experiment has been clarified (material and method section) in the revised manuscript.

Comment #6:

Where can we find the optimization described in the abstract? The optimization process should give information about minimization or maximization value of the response variables.

Response:

Many thanks for the comment.

This point has been treated in the abstract and the following paragraph has been added.

“The optimization process shows that the optimum parameters to attain the highest mechanical properties in terms of hardness and tensile strength at the lowest surface roughness and machine torque are travel speed (TS) of 200 mm/min using cylindrical (Cy) pin geometry at the constant rotation speed (RS) of 600 rpm.”